# CBCT Radiological Features as Predictors of Nerve Injuries in Third Molar Extractions: Multicenter Prospective Study on a Northeastern Italian Population

**DOI:** 10.3390/dj9020023

**Published:** 2021-02-21

**Authors:** Stefano Bigagnoli, Christian Greco, Fulvia Costantinides, Davide Porrelli, Lorenzo Bevilacqua, Michele Maglione

**Affiliations:** 1School of Specialization in Oral Surgery, Unit of Oral Surgery, Department of Medical, Surgical and Health Sciences, University of Trieste, Piazza dell’Ospitale 1, 34129 Trieste, Italy; bigagnolistefano@gmail.com (S.B.); F.COSTANTINIDES@fmc.units.it (F.C.); m.maglione@fmc.units.it (M.M.); 2Unit of Dentistry, Merano Hospital, via Giacomo Rossini 5, 39012 Merano, Italy; christian.greco@asbmeran-o.it; 3Department of Medical, Surgical and Health Sciences, University of Trieste, Piazza dell’Ospitale 1, 34129 Trieste, Italy; dporrelli@units.it

**Keywords:** CBCT, third molars, classification, nerve injures

## Abstract

Background: Neurological alterations are one of the main complications occurring after the third molar extractions. The aim of this prospective multicenter cohort study was to find out Cone Beam Computed Tomography (CBCT) features and distribution of neurological complications in patients undergoing lower third molar surgery and to determine the radiological and patient-related factors that could be correlated to the occurrence of inferior alveolar and lingual nerves injury. Material and Methods: 378 patients who underwent lower third molar extraction from March 2018 to March 2019 were included. Clinical and radiological data were collected. CBCT features were recorded following Maglione et al. classification. Symptoms and characteristics of patients who experienced neurological alterations were evaluated. Results: 193 patients needed a second-level radiological exam (CBCT). In these patients, the most common feature was Maglione class 3: a higher frequency of apical or buccal mandibular canals in direct contact with the tooth was observed. 3.17% of the patients developed a neurological complication. Maglione class 4, increased age, and operative time were all positively correlated with neurological alterations. Conclusions: while the buccal or apical position of the mandibular canal was the more common findings, the lingual position was found to have a higher correlation with a negative outcome. Age and operative time were also found to be risk factors for developing nerve injury in the considered population.

## 1. Introduction

Neurological alterations after third molar extractions are one of the main concerns among oral surgeons. Third molars are indeed positioned closely, and sometimes almost in contact, to at least three branches of the mandibular nerve: lingual nerve (LN), inferior alveolar nerve (IAN), and buccal nerve (BN) are sensitive branches that can be involved in a third molar extraction. The spatial relation of the third molar crowns and roots represents a major challenge for the surgeon, who must take care of these important structures in all surgical procedures. An injury to these branches can result in an array of clinical presentations, from a temporary mild numbness of a small area to a permanent great painful discomfort of a wide anatomical region. This is mainly because these nerves can be injured in many ways, from postoperative swelling compression to cutting bur shearing [1].

Preoperative 3D imaging techniques represent a staple in surgery whenever the surgeon has a suspect of an intimate anatomic relationship of the nerve with the surgical site [2]. For the determination of LN position, it is possible to take advantage of MRI technologies, although they do not represent a routine first-level imaging technique [3].

LN is found to be closely bound to the lingual cortical plate in 20 to 60% of cases [4,5,6,7,8] and above the bone crest in a considerable percentage of cases (4.6 to 21%) [4,5,7,9]. It lays behind the retro-molar region in 0.5 to 1.5% of cases [5,7]. Age and crest atrophy seems to worsen this anatomical relationship [4].

An LN injury gives the greatest discomfort to the patient that often leads to legal consequences [10,11,12,13].

Some characteristics and procedures that can have an impact on the frequency of LN injury are reported to be a too distal lingual incision, too apical sutures, advanced age, total inclusion, the distal inclination of the tooth, lingual split techniques and distolingual ostectomy procedures [14,15,16,17,18].

The mandibular canal, where IAN runs parallel with the mandibular artery and vein, has very often an intimate relationship with the third molar root, with frequent modification in caliber, shape and corticalization in correspondence with the tooth.

IAN be accurately visualized through Computed Tomography (CT) or Cone Beam Computed Tomography (CBCT imaging [2], which can provide surgeons important additional information. This information appears to have a positive impact on temporary injury incidence but not on the frequency of permanent alteration [19]. Some authors report an apical or buccal course of the canal to be more prevalent in some populations [20,21,22,23,24,25]. Other studies report a lingual course to be more common [16,26]. Interradicular canals occur in 4 to 15% of the cases [20,21,27]. Several authors report that in most of the cases, IAN is not in direct contact with the tooth [21,22], although other works point out that a 2D radiographic analysis is not appropriate to claim this observation [28,29,30,31,32,33].

Rood’s signs [34] are commonly used as indicators to proceed with a 3D exam because many studies related them with IAN injuries [30,35,36,37], while others reported neurological complications also without these radiological findings [15,38]. Nowadays, also the minimal superimposition of the canal with the tooth structures is considered to be an indicator of the need for a radiographic examination [2].

One of the main issues that can be visualized with 3D radiographs is the absence of corticalization of the mandibular canal that is in contact with the tooth. This finding is considered by many authors as a risk indicator for IAN injury during third molar surgery [24,39,40,41,42], while others consider it as a mere anatomical feature not related to neurological sequelae [2,38,43].

An increased risk of IAN injuries was reported when a cowbell or tear shape of the canal in contact with the third molar can be visualized [44]. Furthermore, the lingual position of the mandibular canal is considered a risk indicator for sensitivity alteration by some authors [20,26,45].

A new CBCT classification has been recently proposed by Maglione et al. [22]. Seven different classes were edited accordingly to the relative position between the mandibular canal and third molar as they appear in cross-sectional images of a 3D radiograph: as reported in Table 1.

IAN injuries may vary by the entity of discomfort caused [46,47]. Some factors and procedures, believed to be correlated with a higher risk of IAN injuries, include advanced patient age, with a cutoff value of 25 years, some Rood’s signs like root radiotrasparency and deflection in contact with the mandibular canal, intraoperative nerve exposition and lingual split technique application [15,16,31,48,49,50,51]. In addition, general anesthesia seems to be a variable that increases the risk of neurological disturbances after the third molar removal [52,53].

Buccal nerve (BN) runs buccally to the third molar and can be involved when the distal releasing incision is performed. It provides sensitivity for cheek mucosa (and minimally skin), posterior alveolar mucosa, keratinized gingiva, and it can send some terminal branches to the lower and upper lip [54].

However, this is the least investigated structure because its damage has a mild clinical presentation due to conspicuous nerve overlapping in the area [55].

Some authors report a postoperative injury rate that is comparable with the other two branches [56]. Nowadays, the incidence of NL damages seems to be comprised between 0.1% and 22%, while the incidence of IAN injuries during third molar surgery is attested between 0.17% and 8.4% [15].

Some authors report widely higher rates in difficult cases when an anatomical continuity between the canal and the tooth is assessed [57,58].

If sensitive disturbances last more than nine months, with an important cutoff at six months, nerve damage must be considered permanent [1,46,47]. According to various papers, IAN lesions do not recover in a percentage comprised between 0.04% and 0.7% of cases [12,16,59]. LN lesions become permanent in about 1% of cases [16,49]. In order to prevent LN lesion, it seems safe to preoperatively verify the integrity of the lingual cortex, avoiding forces that could break it and, in case of a lingual tilted crown, to perform a horizontal coronectomy in the mesial–distal direction [60,61,62]. If a lingual flap must be elevated, it must be protected with wide and rounded-edge retractors [14,63,64]. It is also mandatory to avoid a too-apical suture [46,60].

In order to minimize IAN injury risk, it seems safe to achieve good surgical access, performing a conservative ostectomy, especially on the distal–lingual side, avoiding high elevation forces using conscious tooth sectioning, and avoiding vigorous socket cleaning [60].

The principal aim of this study was to analyze the distribution of Maglione et al. classification in a population at risk of neurological disturbance after third molar removal. The second aim was to find out if any correlation exists between one of these classes and other perioperative factors with the development of neurological complications in a cohort of Italian patients who underwent third molar extraction.

## 2. Materials and Methods

In this prospective multicenter cohort study, clinical data of 378 consecutive patients who underwent lower third molar extraction were collected over a 1-year time lap (from March 2018 to March 2019).

A total of 212 of these patients were evaluated and treated at the Unit of Oral Surgery, University of Trieste, (Trieste, Italy), whereas 166 were treated at the Dental Service of Merano Hospital (Merano, Bolzano, Italy).

The study was approved by the local ethical committee (UNITS 7/2016). Informed written consent was obtained from each patient to use clinical data for the research that was conducted in agreement with the guidelines of the Helsinki Declaration as revised in 1975 and amended in October 2003. All patients signed an informed consent regarding surgery approval. All surgeries were performed in a standardized fashion, using the same surgical and pharmacological protocols. All surgeons had at least 5 years of experience in third molar removal.

All patients underwent a first level 2D radiological examination (orthopantomography or intraoral X-ray), and if a contiguity relationship between mandibular canal and tooth was suspected, a CBCT was performed, following evidence-based indications of SEDENTEXCT project of the Europe EC, European Commission (2012) [65].

Before surgery, the patient performed a mouth rinse of chlorhexidine 0.12% for one minute. Antibiotic prophylaxis was administered only when there was a risk for infective endocarditis, following the indications of American Heart Association’s guidelines, or for other systemic pathologies related to bacteremia [66].

Surgery was performed under local or general anesthesia. Anyway, 3% mepivacaine with epinephrine 1:100.000 for IAN and NL block plus local infiltration was injected. All sutures were nonabsorbable silk 3.0 that was removed after a week.

After surgery, all patients were administered antibiotic therapy, with amoxicillin 1 g TID for 5 days and an antalgic therapy, with acetaminophen 1000 mg BID for 3 days.

The same operators (S.B. and C.G.) carefully filled in a data collection form for each patient at the end of the intervention. These forms were temporarily stored in a locked facility inside the operative unit. For every patient, a progressive number code was assigned for identification. The following items were collected: gender, date of birth, tooth extracted, presence of CBCT images, Winter and Pell and Gregory (P&G) classification of the tooth, Maglione et al. radiological class [22] (for CBCT only), surgical procedures like flap elevation, osteotomy procedures, tooth sectioning, anesthesia type, operative time (measured from incision to socket cleaning).

Patients were then seen after 7 days for the suture removal and were asked to report any postoperative sensitivity disturbances. In case of a positive answer, the following points were investigated: nerve branch injured, disturbance type, symptoms experienced, an anatomic extension of the sensitivity impairment. Tablets of Alanerv^®^ 920 mg (Alfa Wassermann Spa, Italy) BID for 2 weeks were administered. Then, patients were followed–up every 2 weeks for signs and symptoms monitoring until their resolution. After 6 months without recovery, the neurological complication was considered permanent.

### Statistical Analysis

After the study expected term (1 year), data were gathered from collection forms and put in a spreadsheet (Microsoft Excel, Microsoft Corporation, Redmond, WA, USA). Statistical analysis was performed with the same software.

A *t*-test and Fisher’s exact test were used to assess any difference in sample characteristics between the CBCT patient group and no CBCT group. These tests were also used to evaluate differences between the contact group and no contact group in the CBCT sample. The test was considered significant if α was <0.05.

A chi-squared test was used to evaluate variables distribution inside two populations: patients who suffered a postoperative neurological complication vs. patients who did not. A chi-squared test was also used to analyze the Maglione et al. radiological classes among genders. The test was considered significant if α was <0.05. An ANOVA test with Tukey’s post hoc test was used to analyze the distribution of ages among the Maglione et al. radiological classes.

## 3. Results

The characteristics of patients divided into two groups (those who needed CBCT examination and those who did not) are reported in Table 2.

In the sample of 378 patients, 207 were female (54.76%) and 171 male (45.23%). The average age was 29.38 (±11.97) with a range between 13 and 83 years. Patients were equally distributed in terms of gender and age (differences in the distributions are not statistically significant) in the two groups (CBCT and No CBCT).

A total of 191 (50.53%) third mandibular left molars and 187 (49.47%) third mandibular right molars were extracted.

According to Winter classification, 140 elements (37.04%) were mesio-inclined, 115 (30.42%) were vertical, 67 (17.72%) were horizontal, 33 (8.7%) teeth were disto-inclined and five (1.32%) had an eccentric position. For 17 elements (4.49%), this information did not emerge clearly. As to the relative depth of the element compared to the second molar (P&G classification), 106 teeth (28.04%) lied with the occlusal plane at the level of the second molar (P&G a), 183 (48.41%) had the occlusal plane between the CEJ and the occlusal plane of the second molar (P&G b) and 86 (22.75%) below the CEJ of the second molar (P&G c). For three patients (0.79%), this information was not clear. A total of 101 elements (26.72%) were totally outside the mandibular branch (P&G 1), 232 (61.37%) were partially included in the branch (P&G 2), and 43 (11.37%) were totally included in the branch (P&G 3). For two elements (0.53%), this information was not clear. As expected, The Fisher’s test showed that the impaction of the third molars was lower (1A, 1B, 2A) for patients in which a CBCT was not required (*p* > 0.05), and higher (2B, 2C, 3C) in those which required a CBCT examination.

A total of 369 interventions (97.6%) included a flap elevation, 353 (93.3%) needed ostectomy procedures, and in 308 cases (81.5%), a tooth section was performed. Simple extractions were performed mostly (*p* < 0.001) in patients for whom CBCT was not necessary, while tooth sectioning was performed mostly in the CBCT group (*p* < 0.05), basing the decision also on the evaluation of CBCT images. The average duration of surgery was 31.01 (±16.65) minutes ranging from 3 to 130 min.

For 193 patients (51.06%), a second-level X-ray was required. Table 3 shows the distribution of Maglione et al. radiological classes in the sample. chi-squared test show that the distribution of the radiological classes is similar between males and females (*p* > 0.05), and an ANOVA test with Tukey’s post hoc test shows that the average age distribution is also similar between the classes (*p* > 0.05)

Table 4 reports the distribution of the teeth, of the CBCT group, in the function of their contact with the mandibular canal. Only 29 elements (15.10%) were not in contact with the mandibular canal, and 163 (84.89%) lied in direct contact. A total of 108 mandibular canals (56.25%) were in vestibular or apical position in respect to the tooth, 76 (39.58%) were in lingual position, and 12 (6.25%) run between the roots. IAN position was unknown in only one case in which canal corticalization was absent. For all the positions considered, Fisher’s exact test showed that in most cases, there was a contact between IAN and the third molar (buccal or apical, *p* < 0.001; lingual, *p* < 0.001; interradicular, *p* < 0.05).

Twelve patients (3.17%) showed neurological complications following surgery. In one patient (0.26%), sensitive impairment became permanent. 6 patients (1.58%) suffered IAN complications, 4 (1.06%) LN impairment, one patient (0.26%) developed both IAN and LN paresthesia, and one subject showed LN and BN alterations. The characteristics of the patients with neurological complications are reported in Table 5.

One patient out of 12 (8.3%) had painful symptoms related to the IAN area.

The chi-squared test showed a statistically significant correlation between the lingual position of canals in contact with the tooth (class 4A and 4B, according to Maglione et al.) and the development of postoperative neurological complications. This was true considering both the overall paresthesia sample (*p* = 0.003) and even more significant considering only the IAN paresthesia sample (*p* = 0.001).

A statistically significant correlation emerged between age and the development of postoperative neurological complications. chi-squared test correlated age above 25 with the neurological impairment (*p* = 0.003). Surgical operative times were also shown to be statistically significant (*p* = 0.0034) starting from 30-min duration cutoff. Other variables as pericoronaritis (*p* = 0.78), depth of inclusion (*p* = 0.38), gender (*p* = 0.39), did not show statistical significance. Lack of canal corticalization was also not significant regarding neurological complications (*p* = 0.24).

Furthermore, the absence of a CBCT exam was not significantly correlated with an increased incidence of paresthesia (*p* = 0.60).

## 4. Discussion

The distribution of radiological classes found in the present study does not reflect the general population, in which the extraction of the third molar must be performed but represents a subpopulation in which the orthopantomography revealed an anatomical relationship between IAN and the lower third molar. This sample was per se at risk of postoperative neurological complications.

In this sample, in approximately 85% of patients in which a 3D radiographic examination was required, the lower wisdom teeth were in direct contact with the mandibular canal. In most cases, therefore, a 3D radiographic examination appears to be justified [67].

Most mandibular canals run buccally or apically to the element, while the occurrence of the interradicular canal was rare (6.25%), as reported in the literature [20,21], probably because third molars with three or more roots or in a tilted position are rare.

The incidence of IAN-related lesions following third-molar surgery ranges between 0.17% and 8.4% in the overall population [15], or more (35%) in selected cases in which an intimate tooth-canal relationship is known [57]. Conversely, LN injuries occurrence is considered to lie between 0.1% and 22% [15].

Nerve injury will be permanent in a variable of 0.04% to 0.7% for IAN and around 1% for LN. The persistence of clinical symptoms is related to different nerve injury patterns [1].

In the present study, 3.17% (*n* = 12) of patients reported a neurological complication. Of these, 8.3% (*n* = 1; 0.26% of the total sample) reported an irreversible impairment.

Results showed that the lingual position of the mandibular canal was identified as a significant factor for the development of neurological complications, but only in 4A and 4B subclasses (direct contact with the tooth). Lingual position of the canal, even more, if the latter is placed more coronally, could lead to a compression of the nerve during the tooth elevation procedures as the third molar is mainly approached mesial-buccally and tends to follow a distal–lingual path to leave the socket. According to these results, the authors suggest that coronectomy should be performed before tooth dislocation.

When we considered the 25-year cutoff, as reported in the literature, patient age was a significant factor for postoperative neurological disturbances [15,16,48]. This could be related to a different bone biodynamic and/or to different nerve injury coping patterns at a young age.

Surgical operative times were significantly related to the incidence of neurological complications. It is not clear if the time factor affects postoperative parameters such as edema, or if it is directly proportional to the surgery challenge, or both. However, in this sample, an extended surgery duration can be considered a negative prognostic indicator regarding nerve injuries. Moreover, one of the factors reported to affect surgical time is intraoperative exposure of the IAN [68]. This occurrence, according to some authors, is also connected with an increased risk of postoperative neurological complications [15].

The relationship between the availability of CBTC images and the development of neurological disturbances was not significant. This is consistent with other works [38,43,69]. However, it is clear that radiographic signs, identified on 3D X-ray examination, can provide valuable information, especially in borderline cases such as aberrant positions of the canal, associated pathological structures that shift canal position from the original site, or interradicular mandibular canals. In all these cases, 3D X-ray examination can provide valuable indications for planning the surgery and reduce the risks associated with the extraction of impacted lower third molars [67,70].

The absence of the canal corticalization was not statistically significant. Probably during the surgical maneuvers, a thin cortical bone sheet, like mandibular canal one, can offer very little resistance to compressive crushing elevation forces.

Operator experience was reported to be significant in some works [12,50,71], but this variable was not considered in this study as all surgeons had more than five years of experience.

This study has potential limitations. The first concern is about the limited sample size, resulting in a rare frequency of nerve injuries. Another concern could be about the subjective expression and heterogeneous clinical presentation of nerve lesions, resulting in the need to rely on patient descriptions of symptoms. In this sense, further studies are encouraged.

## 5. Conclusions

Most of the considered third molars belonged to class 3 according to Maglione et al. classification.

Despite the relative rarity of neurological complications, Maglione et al. class 4A or 4B, patients over 25 years old and an extended surgical time (greater than 30 min) can be considered as unfavorable prognostic indicators for the development of sensitivity impairments within the population considered. On the contrary, previous pericoronaritis, the depth of the inclusion, and the gender of the patient determine less important risk factors for postoperative neurological complications.

## Figures and Tables

**Table 1 dentistry-09-00023-t001:** CBCT Radiological Classification by Maglione, Costantinides, Bazzocchi, 2015.

Class	Subtype	Scheme of the RELATIONSHIP between Tooth/Inferior Alveolar Nerve (IAN)
**Class O**: Mandibular canal is not visible on the images (plexiform canal)		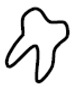
**Class 1**: Mandibular canal runs apically or buccally with respect to the tooth but without touching it (the cortical limitations of the canal are not interrupted)	**1A**: IAN–tooth distance is greater than 2 mm	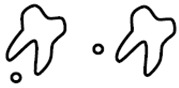
**1B**: IAN–tooth distance is less than 2 mm	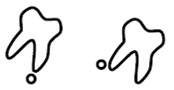
**Class 2**: Mandibular canal runs lingually with respect to the tooth but without touching it (the cortical limitations of the canal are not interrupted)	**2A**: IAN–tooth distance is greater than 2 mm	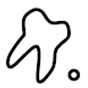
**2B**: IAN–tooth distance is less than 2 mm	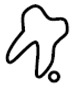
**Class 3**: Mandibular canal runs apical or buccal, touching the tooth	**3A**: In the point of contact, the mandibular canal shows a preserved diameter	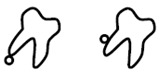
**3B**: In the point of contact, the mandibular canal shows a smaller caliber and/or an interruption of the cortication	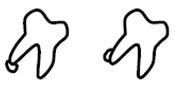
**Class 4**: Mandibular canal runs lingually, touching the tooth	4A: In the point of contact, the mandibular canal shows a preserved diameter	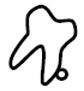
**4B**: In the point of contact, the mandibular canal shows a small caliber and/or an interruption of the cortication	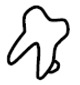
**Class 5**: Mandibular canal runs between the roots but without touching them	**5A**: IAN–tooth distance is greater than 2 mm	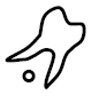
**5B**: IAN–tooth distance is less than 2 mm	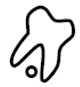
**Class 6**: Mandibular canal runs between the roots touching them	**6A**: In the point of contact, the mandibular canal shows a preserved diameter	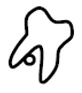
**6B**: In the point of contact, the mandibular canal shows a small caliber and/or an interruption of thecortication	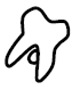
**Class 7**: Mandibular canal runs between fused roots		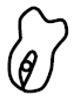

**Table 2 dentistry-09-00023-t002:** Characteristics of patients with Cone Beam Computed Tomography (CBCT) vs. patients with 2D X-rays only (Orthopantomography, OPT).

	CBCT (*n* = 193)	No CBCT ( *n*= 185)	*p* Value
Age mean ± SD	28.06 ± 10.98	30.71 ± 13.43	NS ^a^
**Gender *n* (%)**			
Males	94 (48.7%)	108 (58.4%)	NS ^b^
Females	99 (51.3%)	77 (41.6%) *	NS ^b^
**Pell and Gregory Classification, *n* (%)**			
1-A	14 (7.25%)	42 (22.7%)	<0.05 ^b^
1-B	15 (7.77%)	20 (10.81%)	<0.05 ^b^
1-C	6 (3.1%)	5 (2.7%)	NS ^b^
2-A	19 (9.84%)	30 (16.21%)	<0.05 ^b^
2-B	79 (40.93%)	56 (30.27%)	<0.05 ^b^
2-C	34 (17.61%)	14 (7.56%)	<0.05 ^b^
3-A	1 (0.52%)	1 (0.54%)	NS ^b^
3-B	8 (4.14%)	7 (3.78%)	NS ^b^
3-C	17 (8.81%)	10 (5.4%)	<0.05 ^b^
**Surgical Technique, *n* (%)**			
Simple extraction	1 (0.52%)	16 (8.29%)	<0.001 ^b^
Flap	192 (99.48%)	177 (91.71%)	NS ^b^
Ostectomy	187 (96.89%)	166 (86.01%)	NS ^b^
Tooth sectioning	173 (89.63%)	135 (69.95%)	<0.05 ^b^
**Indication for extraction**			
Orthodontic	8 (4.14%)	9 (4.96%)	NS ^b^
Pericoronitis	104 (53.88%)	100 (54.05%)	NS ^b^
Caries of the third molar	17 (8.81%)	38 (20.54%)	<0.05 ^b^
Caries of second molar	6 (3.11%)	3 (1.62%)	<0.05 ^b^
Prosthetic reasons	1 (0.52%)	0 (0%)	-
Periodontitis of second molar	38 (19.69%)	16 (8.64%)	<0.05 ^b^
Cyst	6 (3.11%)	3 (1.62%)	<0.05 ^b^
Oncological reasons	1 (0.52%)	0 (0%)	-
Mandibular fracture	1 (0.52%)	0 (0%)	-
IAN neuritis	1 (0.52%)	0 (0%)	-
Abscess	1 (0.52%)	0 (0%)	-
No pathology	9 (4.66%)	16 (8.64%)	<0.05 ^b^

SD, standard deviation; ^a^
*t*-test; ^b^ Fisher’s exact test; * significant difference between the groups (*p* < 0.05).

**Table 3 dentistry-09-00023-t003:** Distribution of radiologic classes according to Maglione et al. 2015, in the sample of Cone Beam Computed Tomography (CBCT) patients.

CBCT Radiologic Classes	Frequency in CBCT Patients (*n*,%)	Frequency in Males (*n*,%)	Frequency in Females (*n*,%)	Age (Years, SD)
0	1 (0.52)	0 (0)	1 (1.01)	23, -
1A	6 (3.11)	4 (4.25)	2 (2.02)	31.5 (12.69)
1B	13 (6.73)	7 (7.45)	6 (6.06)	31.46 (11.70)
2A	2 (1.03)	2 (2.13)	0 (0)	34 (5.65)
2B	6 (3.11)	5 (5.32)	1 (1.01)	29.66 (12.66)
3A	37 (19.17)	19 (20.21)	18 (18.18)	25.86 (7.06)
3B	52 (26.94)	24 (25.53)	28 (28.28)	28.09 (12.31)
4A	20 (10,36)	10 (10.64)	10 (10.1)	24.75 (5.71)
4B	47 (24,35)	18 (19.14)	29 (29.29)	29.06 (13.22)
5A	0 (0)	0 (0)	0 (0)	-
5B	2 (1.03)	1 (1.06)	1 (1.01)	33.5 (2.12)
6A	3 (1.55)	3 (3.19)	0 (0)	30.66 (0.57)
6B	4 (2.07)	1 (1.06)	3 (3.03)	28 (8.20)
7	0 (0)	0 (0)	0 (0)	

**Table 4 dentistry-09-00023-t004:** Distribution of the elements regarding direct contact with the mandibular canal *.

Position	No Contact IAN/Third Molar	Contact IAN/Third Molar	Total
Buccal or apical	19	89 ^a^	108
Lingual	8	67 ^a^	76
Interradicular	2	7 ^b^	12
**Total**	29	163 ^a^	192

^a^ significant difference in respect with the “no contact” group (Fisher’s exact test, *p* < 0.001); ^b^ significant difference in respect with “No contact” group (Fisher’s exact test, *p* < 0.05); * one tooth not classifiable for the absence of the canal corticalization (class 0).

**Table 5 dentistry-09-00023-t005:** Characteristics of the patients who developed neurological complications.

*n*	Gender	Tooth	Age	Extraction Indication	Winter Classification	Pell and Gregory Classification	Maglione et al. Classification	Anesthetic Modality	Type of Surgery	Injured Nerve	Duration of Sensory Disturbance	Surgical Time (min)
1	M	48	28	Periodontitis	B	2-C	4A	Local	F/OS/TS	IAN	Permanent	20
2	M	38	35	Periodontitis	M	1-A	4B	Local	F/OS/TS	IAN	Temporary	60
3	F	48	55	Neuritis	D	3-C	4B	Local	F/OS/TS	IAN + LN	Temporary	45
4	F	38	28	Pericoronitis	B	2-B	4B	Local	F/OS/TS	IAN	Temporary	40
5	F	48	40	Pericoronitis	M	2-B	-	Local	F/OS/TS	IAN	Temporary	45
6	M	48	55	Pericoronitis	D	2-B	-	Local	F/OS/TS	IAN	Temporary	35
7	F	38	28	Pericoronitis	O	3-B	4B	Local	F/OS/TS	IAN	Temporary	35
8	F	48	50	Caries	O	2-A	-	Local	F/OS/TS	LN	Temporary	33
9	F	48	26	Pericoronitis	M	2-B	-	Local	F/OS/TS	LN + BN	Temporary	40
10	M	38	32	Cyst	B	2-C	5B	General	F/OS/TS	LN	Temporary	45
11	F	38	23	Pericorontis	B	3-C	4A	General	F/OS/TS	LN	Temporary	20
12	F	48	32	Pericorontis	B	1-A	-	Local	F/OS	LN	Temporary	25

F, flap; OS, ostectomy.; TS, tooth sectioning; B, buccal version; M, mesial version; D, distal version. IAN, inferior alveolar nerve; LN, lingual nerve; BN, buccal nerve.

## Data Availability

Not applicable.

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
