# Peer review of "CBCT Radiological Features as Predictors of Nerve Injuries in Third Molar Extractions: Multicenter Prospective Study on a Northeastern Italian Population"

_dentistry, 2021, doi:10.3390/dj9020023_

Round 1

Reviewer 1 Report

Dear authors,

According to my peer review, the following manuscript entitled - CBCT radiological features as predictors of nerve injuries in third molar extractions: multicenter prospective study on a northeastern Italian Population, addresses a pertinent issue in Oral Surgery and falls within the scope of Dentistry Journal.

Considering its content and presentation, in my opinion, this manuscript may have the merit to be accepted for publication after a minor review.

Please consider these comments in order to improve or clarify your manuscript.

1) How were the patients recruited for your study?It is said that "212 were treated at the unit of Oral Surgery, University of Trieste...and 166 at the Dental Service of Merano Hospital. Did they represent all the patients treated (third molar extraction) in both centers from March 2018 to March 2019?

2) Manuscript text should be abbreviated in some parts, since it repeats tables contents. (for example from line 215 to 224, it is repeated the reasons for third molar extraction previously pointed in the table (Table 1)

3)  By recognizing the loss of extrinsic validity, since the population described does not reflect the general population (line 293), authors should give more emphasis to the rationale of this study.

3.1) How "the suspicion of anatomical continuity between IAN and third molar" was raised? Based on consecutive observations? A scientific terminology is missing and should be considered to replace the idea of "suspicion".

Discussion is well written, however, it should better integrate/express author's findings among the best available evidence.

Author Response

REBUTTAL LETTER

Manuscript: Dentistry journal

CBCT radiological features as predictors of nerve injuries in third molar extractions: multicenter prospective study on a northeastern Italian population

Reviewer(s)' Comments to Author:

Our response: We thank the reviewers for the accurate review of our paper.

Reviewer 1 report:

According to my peer review, the following manuscript entitled - CBCT radiological features as predictors of nerve injuries in third molar extractions: multicenter prospective study on a northeastern Italian Population, addresses a pertinent issue in Oral Surgery and falls within the scope of Dentistry Journal.

Considering its content and presentation, in my opinion, this manuscript may have the merit to be accepted for publication after a minor review.

Please consider these comments in order to improve or clarify your manuscript.

1) How were the patients recruited for your study?It is said that "212 were treated at the unit of Oral Surgery, University of Trieste...and 166 at the Dental Service of Merano Hospital. Did they represent all the patients treated (third molar extraction) in both centers from March 2018 to March 2019?

Our response: We changed the sentence (line 155-156 in the revised manuscript). Now we hope that the method of patient recruitment is clearer.

2) Manuscript text should be abbreviated in some parts, since it repeats tables contents. (for example from line 215 to 224, it is repeated the reasons for third molar extraction previously pointed in the table (Table 1)

Our response: As suggested, some parts of the manuscript were removed to simplify it (line 91-113 and line 225-236 in the revised manuscript).

3)  By recognizing the loss of extrinsic validity, since the population described does not reflect the general population (line 293), authors should give more emphasis to the rationale of this study.

3.1) How "the suspicion of anatomical continuity between IAN and third molar" was raised? Based on consecutive observations? A scientific terminology is missing and should be considered to replace the idea of "suspicion".

Our response: The manuscript was revised as suggested paying attention to the rationale and the scientific terminology

Discussion is well written, however, it should better integrate/express author's findings among the best available evidence.

Our response: We integrated some clinical recommendation and modified discussion and conclusion. (line 331-332, line 350-352, line 370-372 in the revised manuscript)

Reviewer 2 Report

First of all, allow me to congratulate the authors for the relevant subject and important topic to be studied, as it may provide significant information on the field.
This paper fits within the scope of the Journal.

I have some considerations and questions I would like to see answered if possible. 

Probably due to a formatting error, several words appear with a hyphen where there should be none.

Line 17 – instead of “inferior” it should be written “lower”. This error is spread all over the paper. Please correct it. Probably derives from a literal translation of another language into English.

Line 24 – “with a more frequency” replace “more” by, for example, “higher”

Line 28 – “while the buccal or apical position of man- dibular canal was the more common find-ing”. I suggest “while buccal or apical positioning of the mandibular canal was the most common finding”.

Line 38 – “with third molars crowns”. I suggest “with third molar crowns”

Line 46 – “take advance of MRI”. I suggest “take advantage of MRI”

Line 48 – “bounded”. I suggest “bound”

Line 62 – “These information seem to have a positive”. I suggest “This information appears to have a positive

Line 75 – “corticalization of mandibular”. I suggest “corticalization of the mandibular

Line 79 – “It was reported an augmented risk of IAN injuries”. I suggest ““An augmented risk of IAN injuries was reported

Line 80 – “position of man-“. I suggest ““position of the man-

Line 149 – “extraction were collected”. I suggest “extraction was collected”.

Line 162 – “intraoral x-ray radiography”. Please. Choose one of the terms.

Line 173 – “1g tid”. Please choose between “t.i.d” or “TID”. The same fot the word “bid” at the end of this same line

Line 186 – again “bid”

Line 190 – “data were gathered”. I suggest “data was gathered”

Line 312 – “even more if the latter runs more coronally”. Difficult to understand.

Line 318 – “to a different nerve”. I suggest “to different nerve

In the text, authors state that patients “were asked to report any post-operative sensitivity disturbances”, How was that evaluated? Was any scale used? If yes, it should be mentioned in the paper.

As a general comment on the paper, I congratulate the authors for the study design and conclusions. The sample seems too small for some of the study objectives, but that is mentioned as a limitation.

Overall, although literature already has similar studies and available data on the same topics, I think the study is relevant and the paper too. The only not so positive point is the limited mastery that the authors seem to have in the English language. I saw many mistakes and pointed some of them out, but the paper needs a full linguistic revision before being eligible for publishing.

I would like to ask authors if and how the sample size was calculated.

I would also like the authors to comment on why there is no null hypothesis in the paper.

In terms of literature, I think the authors should have included some more recent papers. There is some information coming out on the theme and it should be mentioned. Here are some examples:

- Wang D, Lin T, Wang Y, Sun C, Yang L, Jiang H, Cheng J. Radiographic features of anatomic relationship between impacted third molar and inferior alveolar canal on coronal CBCT images: risk factors for nerve injury after tooth extraction. Arch Med Sci. 2018 Apr;14(3):532-540. doi: 10.5114/aoms.2016.58842. Epub 2016 Mar 23. PMID: 29765439; PMCID: PMC5949900.

- Bozkurt P, Görürgöz C. Detecting direct inferior alveolar nerve - Third molar contact and canal decorticalization by cone-beam computed tomography to predict postoperative sensory impairment. J Stomatol Oral Maxillofac Surg. 2020 Jun;121(3):259-263. doi: 10.1016/j.jormas.2019.07.004. Epub 2019 Jul 18. PMID: 31325623.

- de Toledo Telles-Araújo G, Peralta-Mamani M, Caminha RDG, de Fatima Moraes-da-Silva A, Rubira CMF, Honório HM, Rubira-Bullen IRF. CBCT does not reduce neurosensory disturbances after third molar removal compared to panoramic radiography: a systematic review and meta-analysis. Clin Oral Investig. 2020 Mar;24(3):1137-1149. doi: 10.1007/s00784-020-03231-6. Epub 2020 Feb 12. PMID: 32052178.

- Qi W, Lei J, Liu YN, Li JN, Pan J, Yu GY. Evaluating the risk of post-extraction inferior alveolar nerve injury through the relative position of the lower third molar root and inferior alveolar canal. Int J Oral Maxillofac Surg. 2019 Dec;48(12):1577-1583. doi: 10.1016/j.ijom.2019.07.008. Epub 2019 Jul 28. PMID: 31362896.

- Menziletoglu D, Tassoker M, Kubilay-Isik B, Esen A. The assesment of relationship between the angulation of impacted mandibular third molar teeth and the thickness of lingual bone: A prospective clinical study. Med Oral Patol Oral Cir Bucal. 2019 Jan 1;24(1):e130-e135. doi: 10.4317/medoral.22596. PMID: 30573722; PMCID: PMC6344005.

- Pitros P, O'Connor N, Tryfonos A, Lopes V. A systematic review of the complications of high-risk third molar removal and coronectomy: development of a decision tree model and preliminary health economic analysis to assist in treatment planning. Br J Oral Maxillofac Surg. 2020 Nov;58(9):e16-e24. doi: 10.1016/j.bjoms.2020.07.015. Epub 2020 Aug 14. PMID: 32800608.

- Al Ali S, Jaber M. Correlation of panoramic high-risk markers with the cone beam CT findings in the preoperative assessment of the mandibular third molars. J Dent Sci. 2020 Mar;15(1):75-83. doi: 10.1016/j.jds.2019.08.006. Epub 2019 Oct 19. PMID: 32257003; PMCID: PMC7109491.

Author Response

REBUTTAL LETTER

Manuscript: Dentistry journal

CBCT radiological features as predictors of nerve injuries in third molar extractions: multicenter prospective study on a northeastern Italian population

Reviewer(s)' Comments to Author:

Our response: We thank the reviewers for the accurate review of our paper.

REVIEWER 2

First of all, allow me to congratulate the authors for the relevant subject and important topic to be studied, as it may provide significant information on the field.

This paper fits within the scope of the Journal.

I have some considerations and questions I would like to see answered if possible.

Probably due to a formatting error, several words appear with a hyphen where there should be none.

Line 17 – instead of “inferior” it should be written “lower”. This error is spread all over the paper. Please correct it. Probably derives from a literal translation of another language into English.

Line 24 – “with a more frequency” replace “more” by, for example, “higher”

Line 28 – “while the buccal or apical position of man- dibular canal was the more common find-ing”. I suggest “while the buccal or apical position of man- dibular canal was the more common find-ing”.

Line 38 – “with third molars crowns”. I suggest “with third molar crowns”

Line 46 – “take advance of MRI”. I suggest “take advantage of MRI”

Line 48 – “bounded”. I suggest “bound”

Line 62 – “These information seem to have a positive”. I suggest “This information appears to have a positive”

Line 75 – “corticalization of mandibular”. I suggest “corticalization of the mandibular”

Line 79 – “It was reported an augmented risk of IAN injuries”. I suggest ““An augmented risk of IAN injuries was reported”

Line 80 – “position of man-“. I suggest ““position of the man-“

Line 149 – “extraction were collected”. I suggest “extraction was collected”

Line 162 – “intraoral x-ray radiography”. Please. Choose one of the terms.

Line 173 – “1g tid”. Please choose between “t.i.d” or “TID”. The same fot the word “bid” at the end of this same line

Line 186 – again “bid”

Line 190 – “data were gathered”. I suggest “data was gathered”

Line 312 – “even more if the latter runs more coronally”. Difficult to understand.

Line 318 – “to a different nerve”. I suggest “to different nerve”

Our response: We thank the reviewer for his advice. We have modified what was requested and carefully revised the English throughout the manuscript

In the text, authors state that patients “were asked to report any post-operative sensitivity disturbances”, How was that evaluated? Was any scale used? If yes, it should be mentioned in the paper.

Our response:  Postoperative assessment was done by two expert oral surgeons (M.M. and L.B.) after one week at the time of suture removal for hypoesthesia, paraesthesia, anaesthesia by questioning about tongue, chin, and lip sensibility and performing neurosensory tests like pain test and 2-point discrimination.

Pain test was performed using a sharp explorer. This test was assumed positive when patients could differentiate between the pain elicited by the pressure of a blunt tip and the pain elicited by the pressure of a sharp explorer.  (0 = normal sensitivity, 1 = decreased sensitivity, and 2 = no sensitivity)

Two-point discrimination test was performed with a pair of calipers opened progressively.  (0 = normal sensitivity; 1 = decreased sensitivity (patients could distinguish between tips only when the calipers were open between 14 and 20 mm); and 2 = no sensitivity).

However, these data were not useful for the discussion of the results here obtained, so they were not included in the manuscript.

As a general comment on the paper, I congratulate the authors for the study design and conclusions. The sample seems too small for some of the study objectives, but that is mentioned as a limitation.

Overall, although literature already has similar studies and available data on the same topics, I think the study is relevant and the paper too. The only not so positive point is the limited mastery that the authors seem to have in the English language. I saw many mistakes and pointed some of them out, but the paper needs a full linguistic revision before being eligible for publishing.

Our response: The English was carefully revised throughout the manuscript

I would like to ask authors if and how the sample size was calculated.

Our response: Considering the type of study here proposed and its aims there are several factors that affect the power analysis and the sample size calculation. Thus we chose to select a sample size of the same magnitude of already published studies (n=378). The two major Randomized controlled trial, Ghaeminia et al. (2015) (Ghaeminia H, Gerlach NL, Hoppenreijs TJ, Kicken M, Dings JP, Borstlap WA, de Haan T, Berga SJ, Meijer GJ, Maal TJ Clinical relevance of cone beam computed tomography in mandibular third molar removal: a multicentre, randomised, controlled trial. J Craniomaxillofac Surg 2015;43(10):2158–2167) and Guerrero et al.(2013) (Guerrero ME, Botetano R, Beltran J, Horner K, Jacobs R. Can preoperative imaging help to predict postoperative outcome after wisdom tooth removal? A randomized controlled trial using panoramic radiography versus cone-beam CT. Clin Oral Investig 2013 18:335–342. https://doi.org/10.1007/s00784-013-0971-x) were based, respectively, on 320 and 256 treated teeth.

I would also like the authors to comment on why there is no null hypothesis in the paper.

Our response: Considering the type of study here performed and presented, and the aims taken into consideration we think that the formulation of a null hypothesis was not applicable in this manuscript, as from the data here collected is not immediate to select and use a p- value to accept or reject a null hypothesis

In terms of literature, I think the authors should have included some more recent papers. There is some information coming out on the theme and it should be mentioned. Here are some examples:

  • Wang D, Lin T, Wang Y, Sun C, Yang L, Jiang H, Cheng J. Radiographic features of anatomic relationship between impacted third molar and inferior alveolar canal on coronal CBCT images: risk factors for nerve injury after tooth extraction. Arch Med Sci. 2018 Apr;14(3):532-540. doi: 10.5114/aoms.2016.58842. Epub 2016 Mar 23. PMID: 29765439; PMCID: PMC5949900.

Our response: This reference is cited in the our manuscript with number 44

- Bozkurt P, Görürgöz C. Detecting direct inferior alveolar nerve - Third molar contact and canal decorticalization by cone-beam computed tomography to predict postoperative sensory impairment. J Stomatol Oral Maxillofac Surg. 2020 Jun;121(3):259-263. doi: 10.1016/j.jormas.2019.07.004. Epub 2019 Jul 18. PMID: 31325623.

- de Toledo Telles-Araújo G, Peralta-Mamani M, Caminha RDG, de Fatima Moraes-da-Silva A, Rubira CMF, Honório HM, Rubira-Bullen IRF. CBCT does not reduce neurosensory disturbances after third molar removal compared to panoramic radiography: a systematic review and meta-analysis. Clin Oral Investig. 2020 Mar;24(3):1137-1149. doi: 10.1007/s00784-020-03231-6. Epub 2020 Feb 12. PMID: 32052178.

- Qi W, Lei J, Liu YN, Li JN, Pan J, Yu GY. Evaluating the risk of post-extraction inferior alveolar nerve injury through the relative position of the lower third molar root and inferior alveolar canal. Int J Oral Maxillofac Surg. 2019 Dec;48(12):1577-1583. doi: 10.1016/j.ijom.2019.07.008. Epub 2019 Jul 28. PMID: 31362896.

- Menziletoglu D, Tassoker M, Kubilay-Isik B, Esen A. The assesment of relationship between the angulation of impacted mandibular third molar teeth and the thickness of lingual bone: A prospective clinical study. Med Oral Patol Oral Cir Bucal. 2019 Jan 1;24(1):e130-e135. doi: 10.4317/medoral.22596. PMID: 30573722; PMCID: PMC6344005.

- Pitros P, O'Connor N, Tryfonos A, Lopes V. A systematic review of the complications of high-risk third molar removal and coronectomy: development of a decision tree model and preliminary health economic analysis to assist in treatment planning. Br J Oral Maxillofac Surg. 2020 Nov;58(9):e16-e24. doi: 10.1016/j.bjoms.2020.07.015. Epub 2020 Aug 14. PMID: 32800608.

  • Al Ali S, Jaber M. Correlation of panoramic high-risk markers with the cone beam CT findings in the preoperative assessment of the mandibular third molars. J Dent Sci. 2020 Mar;15(1):75-83. doi: 10.1016/j.jds.2019.08.006. Epub 2019 Oct 19. PMID: 32257003; PMCID: PMC7109491.

Our response: We added the references as requested

Reviewer 3 Report

Dear authors, The clinical study presented in this manuscript is interesting and easy to read, providing precious clinical informations. We don't have specific modifications to ask, as the overall study seems well presented to us, with deep analysis of clinical cases. We have just one (slight) modification to ask: Could you add some more 'clinical' recommandation-interpretation in your conclusion (i.e would you recommend to use CBCT more often, as 5 of your 12 neurological complications appeared without CBCT evaluation??) Therefore we recommend to accept your manuscript for publication

Author Response

REBUTTAL LETTER

Manuscript: Dentistry journal

CBCT radiological features as predictors of nerve injuries in third molar extractions: multicenter prospective study on a northeastern Italian population

Reviewer(s)' Comments to Author:

Our response: We thank the reviewers for the accurate review of our paper.

REVIEWER 3

Dear authors, The clinical study presented in this manuscript is interesting and easy to read, providing precious clinical informations. We don't have specific modifications to ask, as the overall study seems well presented to us, with deep analysis of clinical cases. We have just one (slight) modification to ask: Could you add some more 'clinical' recommandation-interpretation in your conclusion (i.e would you recommend to use CBCT more often, as 5 of your 12 neurological complications appeared without CBCT evaluation??) Therefore we recommend to accept your manuscript for publication

Our response: We thank the reviewer for considerations. Reviewer 1 also raised the same concerns and we changed the text. We integrated some clinical recommendation and modified discussion and conclusion. (line 331-332, line 350-352, line 370-372 in the revised manuscript)
